



# Evaluation of retrieval methods for planetary boundary layer height based on radiosonde data

Hui Li[1,2], Boming Liu[1*], Xin Ma[1*], Shikuan Jin[1], Yingying Ma[1], Yuefeng Zhao[2], and Wei Gong[3]

[1] State Key Laboratory of Information Engineering in Surveying, Mapping and Remote Sensing (LIESMARS), Wuhan University, Wuhan, China
[2] Shandong Provincial Engineering and Technical Center of Light Manipulations & Shandong Provincial Key Laboratory of Optics and Photonic Device, School of Physics and Electronics, Shandong Normal University, Jinan 250014, China
[3] School of Electronic Information, Wuhan University

*Correspondence to*: Boming Liu (Email: liuboming@whu.edu.cn) and Xin Ma (maxinwhu@whu.edu.cn)

**Abstract.** Radiosonde (RS) is widely used to detect the vertical structures of the planetary boundary layer (PBL), and numerous methods have been proposed for retrieving PBL height (PBLH) from RS data. However, an algorithm that is suitable under all atmospheric conditions does not exist. This study evaluates the performance of four common PBLH algorithms under different thermodynamic stability conditions based on RS data collected from nine sites in January–December 2019. The four RS algorithms are the potential temperature gradient method ($GM_\theta$), relative humidity (RH) gradient method ($GM_{RH}$), parcel method (PM) and Richardson number method (RM). Atmospheric conditions are divided into convective boundary layer (CBL), neutral boundary layer (NBL) and stable boundary layer (SBL) on the basis of the potential temperature profile. Results indicate that SBL is dominant at nighttime, whilst CBL dominates at daytime. Intercomparisons show that PBLH retrieved via RM is typically higher than those retrieved using the other methods under all and SBL conditions. PBLH retrieved using $GM_\theta$ and $GM_{RH}$ is relatively low. PBLH from PM is the lowest under all and SBL classifications, and the highest under CBL and NBL classifications. Moreover, the uncertainty analysis shows that PBLH retrieved using different algorithms is consistent in most cases (more than 80%) under CBL and NBL conditions. By contrast, the consistency of PBLH is less than 60% under SBL condition. The average profiles and standard deviations of wind speed and potential



temperature under consistent and inconsistent conditions indicate that consistent cases are typically accompanied by evident atmospheric stratification, such as a large gradient in the potential temperature profile or a low-level jet in the wind speed profile. These findings indicate that the reliability of the PBLH results retrieved from RS data is affected by the structure of the boundary layer. Overall, $GM_\theta$ and RM are appropriate for CBL condition. $GM_\theta$ and PM are recommended for NBL condition. $GM_\theta$ and $GM_{RH}$ are robust for SBL condition. This comprehensive comparison provides a reference for selecting the appropriate algorithm when retrieving PBLH from RS data.

## 1 Introduction

The planetary boundary layer (PBL) is the lowest layer of the atmosphere. Its vertical structure is of highly significant in the study of the environment and climate (Stull, 1988; Garratt et al., 1982; Guo et al., 2016). The structure of PBL is considerably affected by topography, season and weather (Eresmaa et al., 2006; Guo et al., 2016). PBL height (PBLH) is directly related to the accumulation and diffusion of pollutants, and it can also be used as the input parameter of atmospheric chemical models and weather forecast systems (Liu et al., 2020; Shi et al., 2020). The continuous observation of PBL is conducive to investigating the spatial and temporal distributions of pollutants and further optimising pollution simulation (Liu et al., 2018; Seibert et al., 2000). Therefore, monitoring PBLH is important (Seidel et al., 2010).

In recent years, various instruments have been developed to observe the structure of PBL. These instruments include the radiosonde (RS), radar wind profiler, microwave radiometer, lidar and ceilometer (Emeis et al., 2004; Guo et al., 2016; 2021; Liu et al., 2019; Aryee et al., 2020; Jiang et al., 2020). In accordance with the principle of observation, these instruments can be divided into three categories. RS and microwave radiometers can invert the vertical structure of the boundary layer by detecting atmospheric thermodynamic profiles (e.g. temperature and humidity) (Guo et al., 2016; Zhang et al., 2018). Radar wind profilers can detect the vertical distribution of the atmospheric wind



field and calculate PBLH through the variation of the atmospheric dynamic structure (Liu et al., 2020). Lidars and ceilometers observe the vertical structure of the boundary layer through the extinction properties of aerosols in the boundary layer (Haeffelin et al., 2012; Schween et al., 2014; Liu et al., 2018). These instruments provide us with good tools for observing the boundary layer.

Given its high detection accuracy and strong anti-interference capability, RS has been widely used in detecting PBLH (Seidel et al., 2012). PBLH is traditionally retrieved through the height-resolved observation of RS data, such as the profiles of temperature, humidity and wind speed (Kursinski et al., 1997; Zhang et al., 2018). Retrieval methods include surface-based inversion, relative humidity (RH) gradient method ($GM_{RH}$), potential temperature gradient method ($GM_\theta$), Richardson number

method (RM) and the parcel method (PM) (Seidel et al., 2010; Seidel et al., 2012). Surface-based inversion, which was proposed by Bradley et al. (1993), is a clear indicator of a stable boundary layer; its top can also define PBLH. The maximum level of the vertical potential temperature gradient was determined as the current PBLH, indicating a transition from the lower region with less stable convection to the upper region with more stable convection (Stull, 1988; Garratt, 1994; Oke,

1995). Similarly, the minimum level of the vertical RH gradient was defined by Seidel et al. (2010) as the current PBLH. In RM, the ratio of buoyancy-related turbulence to mechanical shear-related turbulence is calculated to obtain the Richardson number (Ri), which determines the current PBLH as the lowest level when Ri crosses a critical value of 0.25 (Vogelezang and Holtslag, 1996). The basic idea of PM is to follow the dry adiabatic process, starting at the surface with the measured or

expected (maximum) temperature up to its intersection with the temperature profile from the most recent RS data. PM determines the PBLH as the equilibrium level of a hypothetical rising parcel of air representing a thermal (Holzworth, 1964). The aforementioned algorithms enhance the understanding of PBLH inversion from RS data. However, no algorithm is suitable for all atmospheric conditions. In addition, with the application of artificial intelligence (AI) technology to

the boundary layer, RS data are required to provide reliable PBLH results as standard values

(Rieutord et al., 2020). Therefore, evaluating the performance of various algorithms under different atmospheric conditions is important (Krishnamurthy et al., 2020).

In the current study, the performance of four common RS algorithms is evaluated under different thermodynamic stability conditions based on RS data collected from January to December 2019.
Moreover, the reasons for the differences amongst the algorithms under different atmospheric conditions are analysed. Lastly, the optimal processing (OP) flow for the RS data retrieval of PBLH is proposed. The remainder of this paper is organised as follows. Section 2 introduces RS data, the classification technique used in PBLH definition and comparison and the retrieval methods. Section 3 objectively introduces and discusses the results of the study. Finally, conclusions are drawn in
Section 4.

## 2 Materials and data

### 2.1 RS observations

An L-band RS is an active measuring instrument that can provide fine-resolution profiles of temperature, humidity, wind speed and direction (Guo et al., 2009; Zhang et al., 2018). The L-band
RS of the China Meteorological Administration is typically launched two times a day at 0000 and 1200 Coordinated Universal Time (UTC) (Guo et al., 2016). Additional RS is launched at 0600 UTC in the Beijing (54511), Wuhan (57494) and Changsha (57687) sites, mostly to improve the prediction capability of high-impact weather in China (Zhang et al., 2018). Here, nine sites equipped with RS and radar wind profilers were used, as shown in Fig. 1. The name, longitude and latitude, altitude and
other information of each site are provided in Table 1. With the exception of the Urumqi site (51463), which has an altitude of approximately 0.9 km, most of the sites are located in low- and medium-level land. The RS data from the nine sites were obtained from January to December 2019.



PBLH estimates are sensitive to the vertical resolution of RS data (Seidel et al., 2010); thus, considering whether to resample or not is necessary when processing RS data. Following the previous RS data processing in China (Liu and Liang, 2010; Zhang et al., 2018; Su et al., 2020), the original L-band RS data were resampled at an interval of 5 hPa from the second reading. Furthermore, RS data with an adjacent height difference greater than 200 m than the original data were deleted to improve the accuracy of the analysis results. After data screening, the number of samples from each site was approximately 700, as shown in Fig. 1.

*2.2 Classification of thermodynamic stability condition*

In accordance with the thermodynamic stability structure, PBL can be divided into three types: convective boundary layer (CBL), neutral boundary layer (NBL) and stable boundary layer (SBL) (Liu and Liang, 2010; Zhang et al., 2018). CBL refers to atmosphere heated by the ground. Atmospheric turbulence is strong in CBL, and unstable stratification occurs. NBL refers to the neutral stratification of the entire atmosphere from bottom to top. Buoyancy in NBL exerts an extremely weak effect on turbulent motion, and it can be disregarded. SBL is formed via inversion stratification accompanied by ground radiation cooling. It typically occurs at night, and it is also known as the nocturnal boundary layer (Stull, 1988; Zhang et al., 2016; Zhang et al., 2020).

PBL types are classified by calculating the potential temperature difference (PTD) between the fifth and second sample points from the surface (Liu and Liang, 2010; Zhang et al., 2020). The threshold value of PTD is set as 0.1 K. Moreover, SBL has to be determined further by using the third and first sample points from the surface. In particular, if $PTD_{5-2} > 0.1$ K and $PTD_{3-1} > 0$ K, then PBL is identified as SBL; if $PTD_{5-2} < -0.1$ K, then PBL is identified as CBL. Other cases can be regarded as NBL (Zhang et al., 2020). The classification results of the nine selected sites are presented in Fig. 1. For all the sites, SBL is the dominant PBL type (i.e. more than 400 samples). This result is attributed





to the fact that the detection time of RS in China is at night, which is conducive to the formation of SBL (Nieuwstadt, 1984; Poulos et al., 2002).

*2.3 Methodology for estimating PBLH*

In the current study, four common methods are used to retrieve PBLH from RS data: $GM_\theta$, $GM_{RH}$,

PM and RM.

$GM_\theta$ is similar to $GM_{RH}$. They analyse the vertical gradient profile of $\theta$ and RH, find the minimum local peak value that exceeds the threshold value by setting the threshold value, determine the height corresponding to the minimum local peak value as PBLH and set the threshold values of the potential temperature and RH vertical gradients as 0.003 K/m and 0/m, respectively (Seidel et al., 2010; Stull,

1988; Garratt, 1994; Oke, 1995).

In PM, the height of PBL is defined as the height from the adiabatic rising air mass to neutral buoyancy under CBL and NBL classification conditions (Stull, 1988). In accordance with Liu and Liang (2010), PBLH is more difficult to retrieve under SBL than under CBL and NBL conditions. Moreover, SBL turbulence can be generated using two major mechanisms: buoyancy forcing and

15 shear driving. If buoyancy forcing-derived and wind shear-derived PBLH are simultaneously generated, then minimum height is estimated as PBLH for SBL.

RM has been proven to be a reliable method for calculating PBLH (Seidel et al., 2012; Vogelezang and Holtslag, 1996). On the basis of previous studies (Guo et al., 2016), height is estimated as PBLH in the current study when the rib exceeds the critical value of 0.25.

For all the inversion methods, PBLH results are limited within 0.15–3 km to avoid the influences of surface noise and high clouds. In addition, a surface-based temperature inversion layer (TIL) is a clear indicator of SBL, wherein inversion height can define PBLH (Bradley et al., 1993; stull, 1989).



Seibert et al. (2000) indicated that the temperature inversion structure differs from the boundary layer structure assumed by the four methods. Therefore, if TIL is found in a sounding, then the four methods are not evaluated.

## 3 Results and discussion

The frequency of different PBL types was investigated in this section. Moreover, PBLH results obtained using different methods were compared with one another. Then, the reasons for the differences amongst the algorithms under different atmospheric conditions were analysed. Lastly, an OP flow for the RS data inversion of PBLH was proposed.

### 3.1 Frequency of different PBL types

The frequency of different PBL types at the nine selected sites was calculated in accordance with the vertical distribution of potential temperature, as illustrated in Fig. 2. Notably, TIL actually belongs to SBL (Seibert et al., 2000). The four methods are not evaluated when TIL is present; thus, the frequency of TIL is also calculated. For the nine selected sites, TIL and SBL account for more than 60%, and even 90% in Jinan (54727) and Changsha (57687). This result indicates that the
atmosphere is in a stable state in most RS observations. From the perspective of observation time, SBL and TIL dominate at 0000 and 1200 UTC, whilst CBL is dominant at 0600 UTC. This finding is attributed to the influence of sunlight; that is, the high surface heat flux during the day is conducive to the formation of CBL, whilst the low heat flux at night is conducive to the formation of SBL (Nieuwstadt, 1984; Stull, 1988; Poulos et al., 2002). Moreover, SBL and NBL can form under
certain meteorological conditions during the day and even occur in the afternoon (Medeiros et al., 2005). Overall, the proportions of CBL, NBL and SBL are similar across these stations, except in Urumqi (51463) and Sansha (59981). In the Urumqi (51463) site, CBL can account for 20%, even in the absence of daytime (0600 UTC) detection data. CBL is mostly concentrated at 1200 UTC. This





result is attributed to the geographical location of Urumqi, where sunset occurs after 1200 UTC during spring and summer (Guo et al., 2019). In the Sansha (59981) site, which is set up on an island, NBL at 1200 UTC can account for 20%, and TIL is less than 5%. This finding indicates that the boundary layer structure is mostly affected by sea breeze.

## 3.2 Intercomparison of PBLH results

Figure 3 shows the quartile of PBLH and the average PBLH of the four methods in three time intervals each day under the four categories. The sample numbers of CBL, NBL and SBL is 374, 918 and 3340, respectively. PBLH exhibits evident diurnal variation, particularly in the All classification (Fig. 3a). PBLH at noon (0600 UTC) is significantly higher than those at other times (the median is approximately 1 km), whilst PBLH results in the morning (0000 UTC) and evening (1200 UTC) are significantly lower than that at noon. This finding is attributed to the strong solar radiation at noon, causing the boundary layer to develop fully at daytime, whilst weak solar radiation leads to maintaining PBLH at a low level; the average height is approximately 0.5 km (Zhang et al., 2016). This finding is similar to that of Guo et al. (2016), who indicated that PBLH is typically less than 1 km at daytime and less than 0.5 km at night. The comparison of the PBLH results obtained using different methods indicates that the mean PBLH retrieved via RM is typically higher than those retrieved using the other methods under All and SBL classifications, and the mean PBLH retrieved using $GM_\theta$ and $GM_{RH}$ is relatively low. The mean PBLH retrieved using RM is the highest at 0000 and 0600 UTC under CBL and NBL classifications. Moreover, the mean PBLH retrieved using PM is the lowest under All and SBL classifications and the highest under CBL and NBL classifications. Similarly, PM mixing heights are lower than those of the other methods (Seidel et al., 2010).





*3.3 Uncertainty analysis*

Figure 4 presents two case studies of PBLH determination using the four different methods under CBL classification. The first case is at the Beijing (54511) station at 0000 UTC on 10 June 2019 (Figs. 4a–4c). The PBLH results of $GM_\theta$ and $GM_{RH}$ are the same (0.26 km) and similar to those of

PM and RM (0.29 km). From the wind speed and temperature profiles (Fig. 4c), evident low-level jets and uplifted inversion layers are observed. The second case is at the Urumqi (51463) station at 1200 UTC on 05 August 2019 (Figs. 4d–4f). PBLH retrieved using $GM_\theta$ and PM is approximately 2.1 km, which differs from that retrieved using $GM_{RH}$ and RM (approximately 1 km). In the wind speed profile, wind shear appears at the height of the two PBLH results. Figure 5 illustrates the case

studies of PBLH determination under NBL classification at the Qingdao (54857) station at 0000 UTC on 10 June 2019 (Figs. 5a–5c) and at the Beijing (54511) station at 0000 UTC on 15 March 2019 (Figs. 5d–5f). The PBLH results determined using the four methods in the first case are consistent (i.e. approximately 0.25 km), whilst the PBLH results determined using the four methods in the second case are significantly different. PBLH retrieved using $GM_\theta$ and PM is approximately

1.45 km, whilst the results of $GM_{RH}$ and RM are approximately 0.3 km. Similar to the CBL cases, evident uplifted inversion layers are observed in the first case (Fig. 5c). In the second case, the existence of a low-level jet at the height of the two PBLH results is reported. Lastly, the case studies under CBL classification are presented in Fig. 6. The first case is at the Urumqi (51463) station at 0000 UTC on 23 February 2019. The second case is at the Beijing (54511) station at 0000 UTC on

10 November 2019. Similar to the previous cases, evident uplifted inversion layers are noted when PBLH is retrieved using the four methods. These results indicate that the reliability of PBLH results retrieved from RS data is affected by the structure of the boundary layer.

To investigate the effect of the boundary layer structure, we define consistency to evaluate PBLH results obtained using different methods. For each sample, if the heights determined by more than

three methods are similar (i.e. the height difference is less than 0.3 km), then the PBLH results





obtained using these methods are determined to be consistent. Otherwise, the PBLH results are determined to be inconsistent. In addition, if the PBLH result is unavailable, then it is defined as an invalid value (nan). In this manner, we generate statistics on the consistency of all the algorithms under all the classification conditions. The statistical results are provided in Table 2. Under CBL

condition, $GM_\theta$ achieves the highest consistency, accounting for 83.96%, whereas $GM_{RH}$ presents the lowest consistency, accounting for 74.06%. The consistency of the two other methods is approximately 80%. For NBL classification, $GM_\theta$ also exhibits the highest consistency, accounting for 91.72%, whereas RM demonstrates the lowest consistency, accounting for 69.50%. Under SBL condition, the consistency of $GM_\theta$, $GM_{RH}$, PM and RM is 58.35%, 57.87%, 49.34% and 28.47%,

respectively. Moreover, the PBLH results retrieved using different methods are also evaluated under TIL condition. In this classification, the temperature inversion height is regarded as the standard value and compared with the results retrieved using the four methods. The retrieval results of $GM_{RH}$ and $GM_\theta$ exhibit the highest consistency (above 90%). These results indicate that the consistency of PBLH retrieved using $GM_\theta$ and $GM_{RH}$ is higher than those retrieved using other methods. These

findings are consistent with those of Seidel et al. (2010). Simultaneously, under NBL and SBL classifications, the proportion of effective PBLH results for $GM_\theta$, $GM_{RH}$ and PM is extremely high, and the proportion of invalid values (nan) is less than 1%. By contrast, the proportion of invalid values for RM is 18.30% and 63.47% under NBL and SBL classifications, respectively. This finding indicates that $GM_\theta$, $GM_{RH}$ and PM are more effective than RM under NBL and SBL conditions.

Under TIL conditions, 95.95% of the PBLH results from RM are defined as nan. This finding may be attributed to the formation of TIL being frequently related to the radiative cooling of the surface. When TIL occurs, turbulence is weak, and thus, the probability of using RM to retrieve PBLH is small (Seidel et al., 2012).

In accordance with the aforementioned consistency, the average profiles and standard deviations of

the wind speed and potential temperature of consistent and inconsistent cases under CBL, NBL and





SBL classifications are presented in Fig. 7. Under CBL classification, the mean wind speed profile of consistent cases is similar to that of inconsistent cases (Fig. 7a), whilst the mean potential temperature profile of consistent cases has a larger gradient than that of inconsistent cases (Fig. 7d). This phenomenon also occurs in NBL classification (Figs. 7b and 7e). By contrast, the mean wind speed profile of consistent cases differs from that of inconsistent cases under SBL classification (Fig. 7c), and an evident low-level jet (0.3–0.4 km) is observed in the mean wind speed profile of consistent cases. The mean potential temperature profile of consistent cases is in accord with that of inconsistent cases (Fig. 7d). The mean potential temperature profile of consistent cases exhibits an evident gradient under CBL and NBL classifications, and the mean wind speed profile of consistent cases has an apparent low-level jet under SBL classification. These results indicate that consistent cases are typically accompanied by noticeable atmosphere stratification, such as a large gradient in the potential temperature profile or a low-level jet in the wind speed profile. Liu et al. (2020) compared PBLH from RS and a radar wind profiler. They pointed out that the height difference between PBLH from RS and from the radar wind profiler is evident when the vertical structure of the atmosphere presents no evident stratification.

*3.4 Optimisation process*

In accordance with the preceding uncertainty analysis, we can propose the OP flow for PBLH inversion from RS data. For RS data, the first step is to confirm the structure type of the boundary layer on the basis of the potential temperature and temperature profile. The appropriate method is then selected for different types of boundary layer. Considering the effective inversion number and consistency in Table 2, $GM_\theta$ and RM are recommended for use under CBL condition. Under NBL condition, $GM_\theta$ and PM exhibit the highest consistency, and thus, are recommended for use. Under SBL condition, $GM_\theta$ and $GM_{RH}$ exhibit similar performance and are recommended for use. When TIL is present, the height of the temperature inversion top is defined as PBLH.



Figure 8 shows the quartiles and average values of PBLH for each method and OP at the nine selected sites. Here, the OP of RS data is the use of $GM_\theta$ under CBL, NBL and SBL classification conditions, and the temperature inversion height is regarded as PBLH under TIL condition. With the exception of the Sansha (59981) site, PM and RM overestimate PBLH in each site relative to OP,

whilst $GM_\theta$, $GM_{RH}$ and OP have similar PBLH. PBLH determined using OP is in the average level of the four other methods and relatively stable. This finding is consistent with that of Aryee et al. (2020), who indicated that the gradient method is superior to RM and other methods and can produce extremely low deviations and high statistical correlation coefficients. Figure 8 shows evident regional differences in PBLH. The PBLH results of the Urumqi site (51463) in Northwest China and

the Beijing site (54511) in North China are significantly higher than 0.5 km. In particular, the average PBLH of the Urumqi (51463) and Beijing (54511) sites is higher than those of other sites when RM is used, and the number of average PBLH results is 0.98 km and 0.82 km, respectively. In the inland and coastal areas of southeast China, the average PBLH is generally lower than 0.5 km, even in Sanya (59948), where the average PBLH is approximately 0.3–0.4 km. Such regional

differences are due to various reasons, and certain differences exist in the dominant mechanisms of PBL development in various regions. For example, in the Urumqi site (51463) in Northwest China, net radiation is significantly lower than that in the south. This phenomenon is due to the dry climate, which makes the surface latent heat flux caused by evapotranspiration small; thus, most of the heat is transported to the atmosphere through sensible heat, which is conducive to the development of PBL

(Wang and Wang, 2014; Guo et al., 2019). By contrast, high soil moisture can cause a relatively shallower diurnal PBL over the southeast coast (Mcgrath-Spangler and Denning, 2012). This result is consistent with the analysis based on RS data and the reanalysis data from January 2011 to July 2015 of Guo et al. (2016).



## 5 Summary and conclusions

The performance of four common PBLH retrieval algorithms is evaluated under different thermodynamic stability conditions on the basis of the RS data of nine sites in China from January to December 2019. The reasons for the differences amongst the methods under varying atmospheric conditions are analysed. Finally, the OP flow of PBLH retrieval based on RS data is proposed.

In accordance with the vertical distribution of the potential temperature profile, the frequency of different PBL types in the nine selected sites are calculated. The results show that SBL and TIL are dominant, particularly at 0000 and 1200 UTC, whilst CBL is dominant at 0600 UTC. Moreover, by comparing PBLH retrieved using different methods under varying conditions, the mean PBLH retrieved using RM is typically higher than those retrieved using the other methods under All and SBL conditions, and the mean PBLH retrieved using PM is the lowest. By contrast, the mean PBLH retrieved using PM is the highest under CBL and NBL classifications. The mean PBLH retrieved using $GM_\theta$ and $GM_{RH}$ is relatively low. Then, an uncertainty analysis is conducted for the consistent and inconsistent special cases of the four methods under different classification conditions. The results show that under CBL and NBL conditions, PBLH retrieved using different methods is consistent in most cases (more than 80%). By contrast, the consistency of PBLH is less than 60% under SBL condition. $GM_\theta$ exhibits the highest consistency under all conditions, and $GM_{RH}$ and PM are more effective than RM under NBL and SBL conditions. Meanwhile, the average profiles and standard deviations of the wind speed and potential temperature of consistent and inconsistent cases under CBL, NBL and SBL classifications are analysed. The results indicate that consistent cases are typically accompanied by evident atmosphere stratification, such as a large gradient in the potential temperature profile or a low-level jet in the wind speed profile. Finally, the OP flow for the RS data retrieval of PBLH is proposed. $GM_\theta$ and RM are recommended for use under CBL condition. $GM_\theta$ and PM exhibit the highest consistency and are appropriate for NBL condition. $GM_\theta$ and $GM_{RH}$ are

robust for SBL condition. When TIL is present, the height of the temperature inversion top is defined as PBLH.

The results of this study help in understanding the performance of PBLH retrieval methods and the characteristics of PBL in China. It provides a reliable process for inverting boundary layer height results from RS data. Future work will explore the application of AI algorithms to boundary layer inversion.

**Data availability**

The L-band radiosonde data used in this paper can be provided for non-commercial research purposes upon request (Dr. Boming Liu: liuboming@whu.edu.cn).

**Author contributions**

The study was completed with close cooperation between all authors. H. Li and B. Liu conceived the idea of this manuscript; H. Li and B. Liu conducted the data analyses and co-wrote the manuscript; X. Ma, S. Jin, Y. Ma, Y. Zhao and W. Gong discussed the experimental results, and all coauthors helped reviewing the manuscript.

**Competing interests.**

The authors declare that they have no conflict of interest.

**Acknowledgements.**

This work was financially supported by the National Natural Science Foundation of China under grant 42001291, 41801261, 41827801 and Project funded by the China Postdoctoral Science Foundation 2020M682485.



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





**Table 1.** Summary of RS launch locations and durations of soundings from each site.

| Site number | Site | Longitude | Latitude | Elevation (m) | Total Observation Period |
|---|---|---|---|---|---|
| 51463 | Urumqi | 87.7 | 43.8 | 936 | 01/2019—12/2019 |
| 54511 | Beijing | 116.5 | 39.8 | 31 | 01/2019—12/2019 |
| 54727 | Jinan | 117.5 | 36.7 | 264 | 01/2019—12/2019 |
| 54857 | Qingdao | 120.3 | 36.1 | 75 | 01/2019—12/2019 |
| 57494 | Wuhan | 114.1 | 30.6 | 24 | 01/2019—12/2019 |
| 57687 | Changsha | 112.8 | 28.1 | 120 | 01/2019—12/2019 |
| 59758 | Haikou | 110.3 | 20.0 | 65 | 01/2019—12/2019 |
| 59948 | Sanya | 109.6 | 18.2 | 364 | 01/2019—12/2019 |
| 59981 | Sansha | 112.3 | 16.8 | 6 | 01/2019—12/2019 |





**Table 2.** Consistency statistics of algorithms under different classification conditions.

| Type | Consistency | GM$_\theta$ | GM$_{RH}$ | PM | RM |
|------|-------------|-------------|-----------|-----|-----|
| **CBL** | Consistent | 83.96% | 74.06% | 80.48% | 80.75% |
| | Inconsistent | 13.37% | 24.60% | 14.97% | 16.31% |
| | Nan | 02.67% | 01.34% | 04.55% | 02.94% |
| **NBL** | Consistent | 91.72% | 84.97% | 86.93% | 69.50% |
| | Inconsistent | 7.41% | 14.49% | 12.09% | 12.20% |
| | Nan | 00.87% | 00.54% | 00.98% | 18.30% |
| **SBL** | Consistent | 58.35% | 57.87% | 49.34% | 28.47% |
| | Inconsistent | 41.23% | 41.35% | 50.66% | 08.05% |
| | Nan | 00.42% | 00.78% | 00.00% | 63.47% |
| **TIL** | Consistent | 94.85% | 90.40% | 55.58% | 03.07% |
| | Inconsistent | 04.86% | 09.60% | 44.42% | 00.98% |
| | Nan | 00.29% | 00.00% | 00.00% | 95.95% |



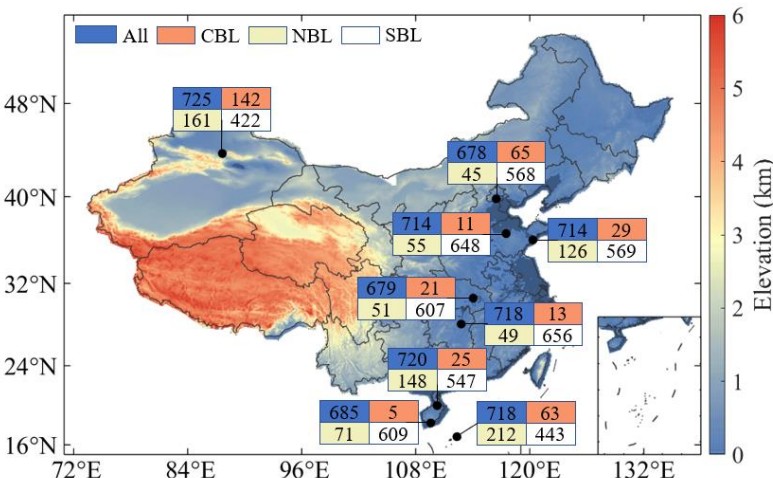

**Figure 1.** Geographic distribution of RS sites (black dots). The text label represents the number of total cases (blue), CBL cases (orange), NBL cases (yellow) and SBL cases (white).





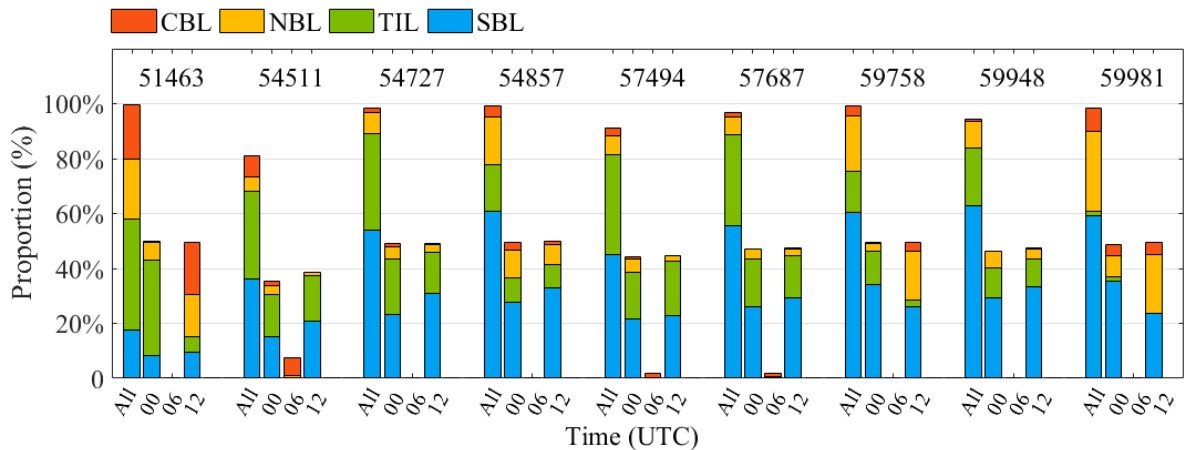

**Figure 2.** Statistics of the classification number of the nine selected sites at different times. The red, orange, green and blue squares represent CBL, NBL, TIL and SBL, respectively.



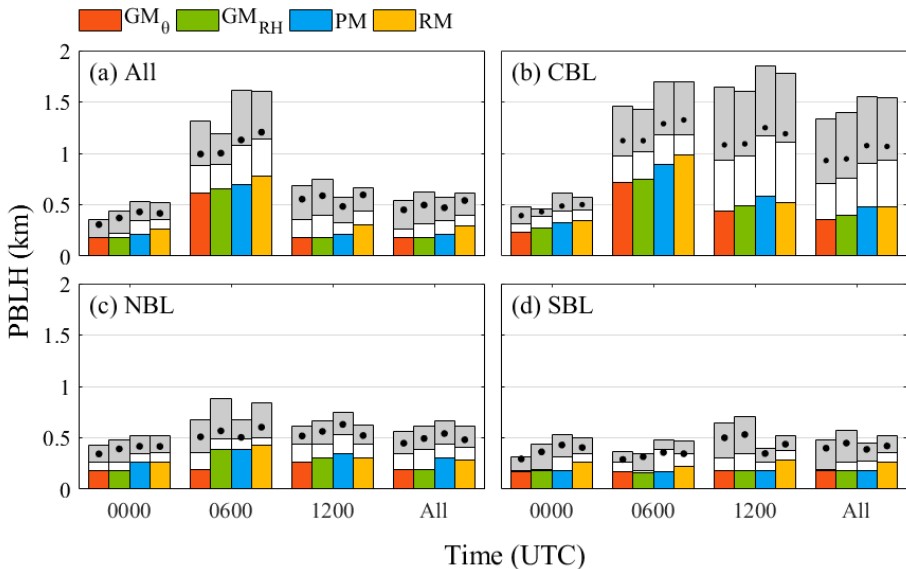

**Figure 3.** Daily quartile and average height of PBLH under (a) all conditions, (b) CBL, (c) NBL and (d) SBL. For each method, the 25th, 50th and 75th percentile values are shown in coloured, white and grey bars, respectively. The solid black dots represent the average PBLH.

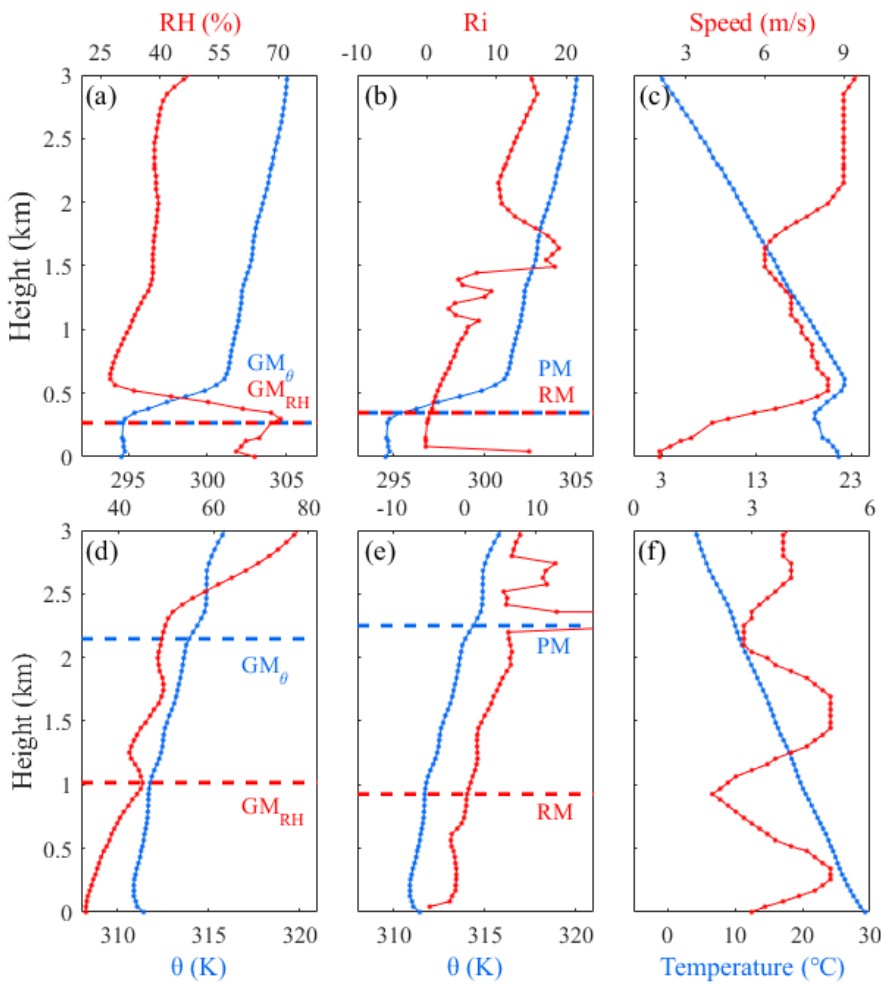

**Figure 4.** Case studies of PBLH determination from (a) GM$_\theta$ (blue) and GM$_{RH}$ (orange), (b) PM (blue) and RM (orange) and (c) profiles of temperature (blue) and wind speed (orange) under CBL classification. (d), (e) and (f) are same as (a), (b) and (c) but in different cases.



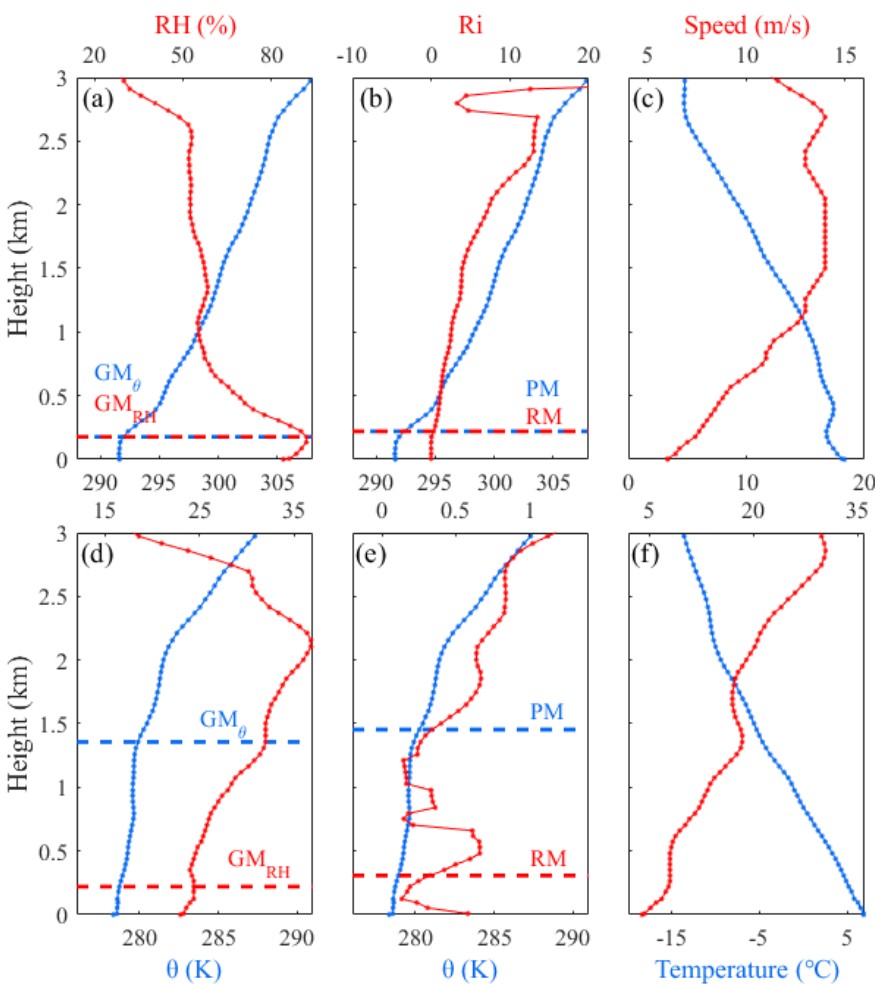

**Figure 5.** Same description as that in Figure 4, but under NBL classification.

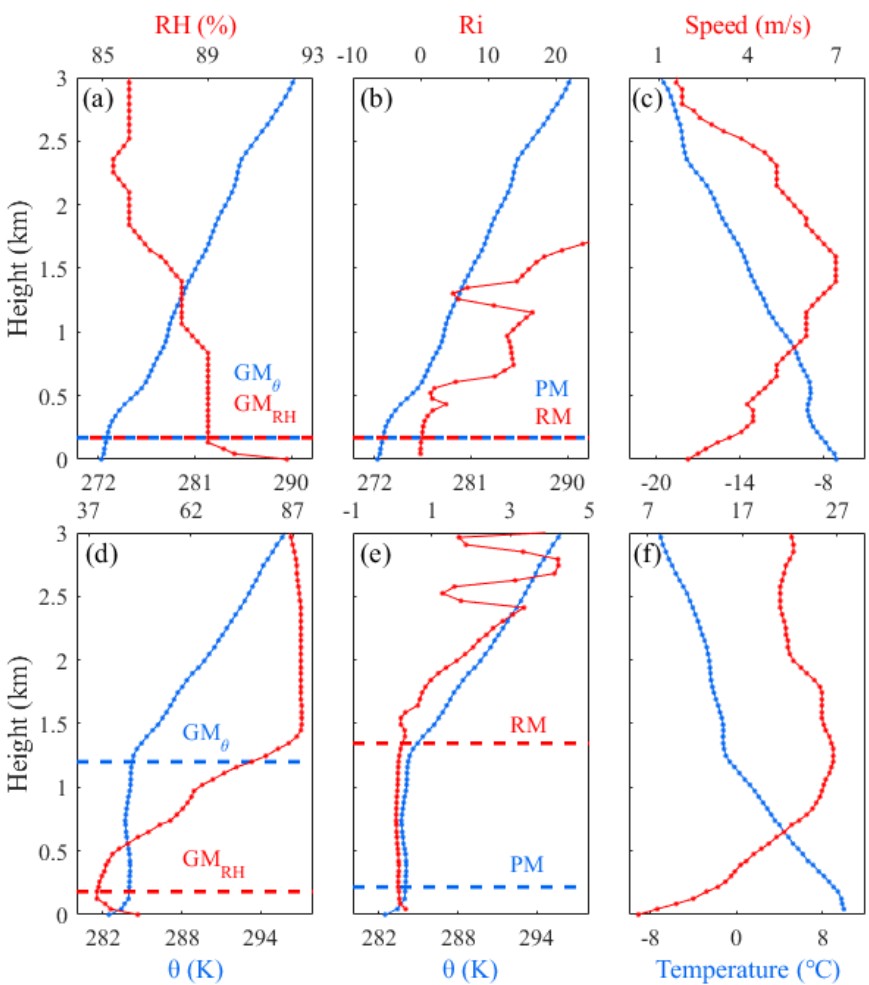

**Figure 6.** Same description as that in Figure 4, but under SBL classification.



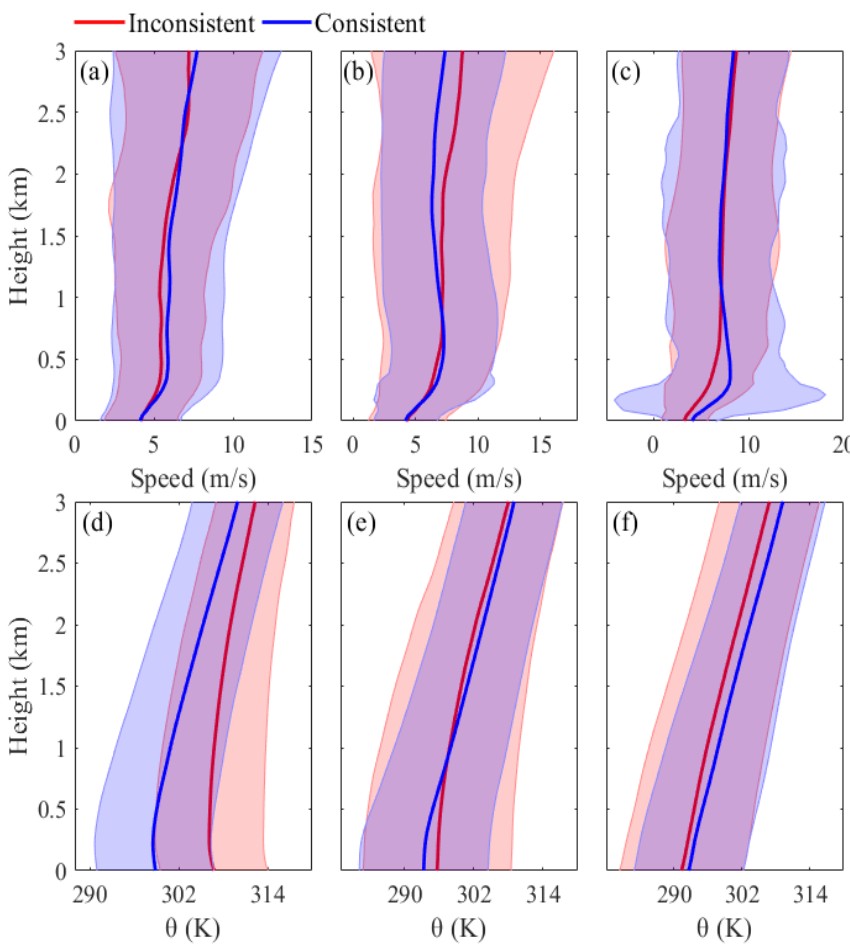

**Figure 7.** Mean (solid line) and standard deviation (shadow) profiles of wind speed and potential temperature in (a, d) CBL, (b, e) NBL and (c, f) SBL classifications.

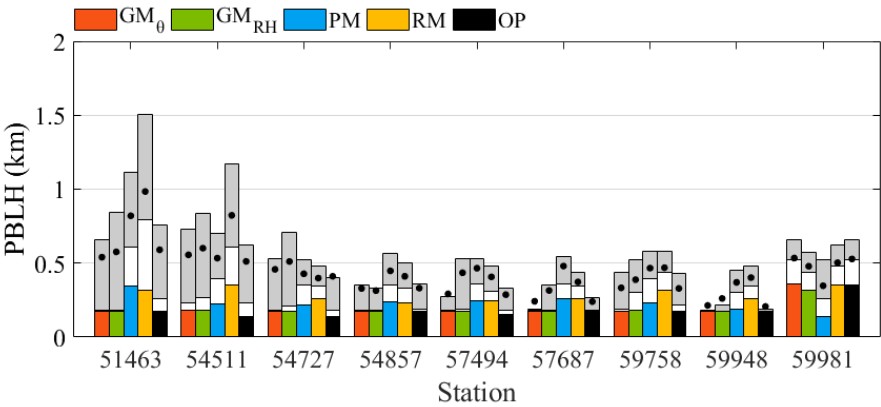

**Figure 8.** Quartile and average heights of PBLH for various methods at the nine selected sites. For each method, the 25th, 50th and 75th percentile values are shown in coloured, white and grey bars, respectively. The solid black dots represent the annual average heights.

