# Peer review of "Evaluation of retrieval methods for planetary boundary layer height based on radiosonde data"

_Atmospheric Measurement Techniques, 2021_

## Author Comment (AC1)

**Response to Reviewer 1's Comments**

***Response: We thank the anonymous reviewer for his/her comprehensive evaluation and thoughtful comments. We have addressed the reviewers' concern one by one. For clarity purpose, here we have listed the reviewer' comments in plain font, followed by our response in bold italics.***

The manuscript described and inter-compared performances of four methods (PM, RM, $GM_{RH}$, and $GM_\theta$) which were widely used to estimate PBLH from RS data. In general, the study fit the scope of the journal and the manuscript was well organized. These results shown in the manuscript can be regarded as a useful reference when selecting boundary layer algorithms. Additionally, there are still several points that need to be clarified before it could be considered for acceptance.

***Response: Thanks for the reviewer's positive comments on our manuscript.***

1. In Abstract section, some sentences are not clear enough between Line 19 and Line 25 of Page 1, such as "PBLH from PM is the lowest under all and SBL classifications, and the highest under CBL and NBL classifications". Please rephrase or clarify.

***Response: We gratefully appreciate for your valuable comment. In Abstract section, we rephrase the following paragraph:***

***"Results indicate that SBL is dominant at nighttime, whilst CBL dominates at daytime. Under all and SBL classifications, PBLH retrieved by RM is typically higher than those retrieved using the other methods. By the contrary, PBLH result retrieved by PM is the lowest. Under CBL and NBL classifications, PBLH retrieved by PM is the highest. PBLH retrieved by $GM\theta$ and $GMRH$ is relatively low under all classifications. Moreover, the uncertainty analysis shows that the consistency of PBLH retrieved by different algorithms is more than 80% under CBL and NBL classifications. By contrast, the consistency of PBLH is less than 60% under SBL classification."***

2. The study focused on estimating the performances of four PBLH calculation methods. But, in the Introduction section, the description of the advance of the subject (namely, the various comparison and estimation of PBLH methods in existing research) is not sufficient enough.

*Response: Per your kind suggestion. We have made a further comparative study on the existing PBLH inversion methods, which have been added to the Introduction section in the revised manuscript.*

3. How much data were used in the study obtained from sites of Beijing, Wuhan, Changsha at 0600 UTC? Since the 0600 UTC is afternoon at local, there should be more CBL and NBL cases in these three cities (as shown in Figure 1 and Page 6 Line 1-2).

*Response: We gratefully appreciate for your valuable comment. In this study, the RS data of Beijing, Wuhan and Changsha sites at 0600 UTC accounted for 13.17%, 2.15% and 2.16% of the data of their respective sites respectively. The proportion of RS data at 0600 UTC is very small, except at 0600 UTC, where CBL is dominant, and from other times as well as the overall time, SBL is still dominant, as shown in Figure 2.*

4. In addition to the TIL, have any additional indicators been added to filter data in order to remove cases under extreme weather conditions? Will the extreme weather, such as rain, snow, fog and storms, impact on the estimation of the boundary layer for RS data?

*Response: We gratefully appreciate for your valuable comment. This study focuses on comparing the performance of the retrieval methods. The presence or absence of cloud rejection and some weather extremes do not have a significant impact on the evaluation process in this study, as the evaluation is carried out under the same conditions. Zhang et al. (2020) compared the mean PBLHs under clear and cloudy conditions and found that the diurnal variation of PBLH is stronger under clear conditions than under cloudy conditions, but the trend of PBLH variation was consistent and the difference in PBLH was not significant.*

*Reference*

*Zhang, Y., Sun, K., Gao, Z., Pan, Z., Shook, M.A., and Li, D.: Diurnal Climatology of Planetary Boundary Layer Height Over the Contiguous United States Derived From AMDAR and Reanalysis Data, Journal of Geophysical Research: Atmospheres, 125, https://doi.org/10.1029/2020jd032803, 2020.*

5. Whether the performance estimations and the OP method are affected by geographic location? I noticed that the mean value of PBLH obtained by OP method are lowest or nearly lowest among the four methods in some cities, such as 57494, 57687, 59758, and 59948.

*Response: We gratefully appreciate for your valuable comment. The performance estimations and the OP method are valid for all geographic locations. The reason why the mean value of PBLH obtained by OP method are lowest or nearly lowest among the four methods in some cities is that in OP method, when TIL is present, the height of the temperature inversion top is defined as PBLH, but the other four methods do not take TIL into account. The PBLH of TIL is generally low, and the proportion of TIL in each time period is relatively large in the four cities of 57494, 57687, 59758 and 59948, as shown in Figure 2.*

6. It is better to add a flow chart for the OP method as described in section of 3.4.

*Response: Per your kind suggestion. We show a flow chart of OP method, as shown below. Due to this process is very simple, we did not add a picture in the paper.*

[Figure]

*Figure S1. Flow chart of OP method.*

---

## Author Comment (AC2)

**Response to Reviewer 2's Comments**

*Response: We thank the anonymous reviewer for his/her comprehensive evaluation and thoughtful comments. We have addressed the reviewers' concern one by one. For clarity purpose, here we have listed the reviewer' comments in plain font, followed by our response in bold italics.*

This study focuses on the performance of four common RS methods under different thermodynamic stability conditions and proposes an optimal processing flow for the RS data retrieval of PBLH. This study provides an optimal RS standard value retrieval method for further inversion of the PBLH through artificial intelligence algorithms. The article has a clear structure and contributes to this area. But in my opinion, there are still some problems should be solved before publishing.

*Response: Thanks for the reviewer's positive comments on our manuscript.*

1. For gradient method, the author mentioned that the threshold values of the potential temperature and RH vertical gradients were set as 0.003 K/m and 0/m, respectively. Why need to set this threshold? Similarly, why was the threshold of Ri set to 0.25, and whether the sensitivity test has been carried out?

*Response: We gratefully appreciate for your valuable comment. Taking the case of May 25, 2019 as an example, as shown in Figure 1 of the response, Figs. 1a and 1c is a vertical gradient image of potential temperature and RH with no threshold set. It can be seen from the Figs. 1a and 1c that there is an oscillation curve before the local peak of the vertical gradient occurs, which affects the search for the local peak of the vertical gradient. Therefore, we have found the most appropriate threshold of 0.003K/m and 0/m through many experiments, which can eliminate the influence of oscillation on the final result, as shown in Figs. 1b and 1d.*

[Figure]

*Figure 1. Vertical gradients of the potential temperature and RH.*

*The reason why the threshold of Ri set to 0.25 is that Guo et al. (2016) have carried out sensitivity analysis on the threshold of Ri and indicated that it is appropriate to set it to 0.25.*

*Reference*

*Guo, J., Miao, Y., Zhang, Y., Liu, H., Li, Z., Zhang, W., He, J., Lou, M., Yan, Y., and Bian, L.: The climatology of planetary boundary layer height in China derived from radiosonde and reanalysis data, Atmospheric Chemistry and Physics, 16, 13309-13319, https://doi.org/10.5194/acp-16-13309-2016, 2016.*

2. About the RS data, why did author choose only nine sites for the experiment? Will there be similar results using data from all sites in the country?

*Response: We gratefully appreciate for your valuable comment. The nine sites in China were selected because they are equipped with radar wind profiler, which facilitates the subsequent comparison of RS observations with radar wind profiler observations for verification. For the second question, Guo et al. (2021) investigated the global boundary layer height using high-resolution RS data and reanalysis data, which included about 120 stations from China, and the results of the spatial distribution and diurnal variation of the boundary layer height were similar to our study.*

*Reference*

*Guo, J., Zhang, J., Yang, K., Liao, H., Zhang, S., Huang, K., Lv, Y., Shao, J., Yu, T., and Tong, B.: Investigation of near-global daytime boundary layer height using high-resolution radiosondes: First results and comparison with ERA-5, MERRA-2, JRA-55, and NCEP-2 reanalyses, Atmospheric Chemistry and Physics Discussions, 1-39, https://doi.org/10.5194/acp-2021-257, 2021.*

3. Section 1: In the third part of the introduction, only a few methods compared in this paper are described in the description of the existing RS data retrieval methods of PBLH, which should be described more comprehensively.

*Response: Per your kind suggestion. We have made a further comparative study on the existing RS data retrieval methods of PBLH, which have been added to the Introduction section in the revised manuscript.*

4. P4-L18-19: This article studies PBLH retrieval method based on RS data. What is the role of radar wind profiler here?

*Response: We gratefully appreciate for your valuable comment. As in the answer to the second question, the purpose of adding the radar wind profile here is to show that the nine sites selected for this study are equipped with a radar wind profiler, so that the RS data can be compared with the radar wind profiler for verification of the boundary layer height observations in the future.*

5. P6-L6: Although there is an explanation for the abbreviation of $GM_\theta$ in the abstract, what does the $\theta$ refers to here, should be explained again.

*Response: Per your kind suggestion. We have added an explanation of the meaning of $\theta$ in the revised manuscript*

6. P6-L19: Does the "rib" here refer to the Richardson number? If so, "b" should be in the form of subscript. And the specific meaning of $ri_b$ needs to be explained.

*Response: Per your kind suggestion. The "rib" in this context does refer to the Richardson number. We have changed "b" to the standard subscript form according to your suggestion and explained it in detail*

7. Figure 2: Why is the proportion of various categories not 100% at All time? At different times, the proportion of different categories is relative to all the cases of a site, or relative to the effective cases?

*Response: We gratefully appreciate for your valuable comment. The reason why the total percentage of each category at all times is not 100% is that we have excluded some cases with abnormal data and not all cases are valid. The proportion of different categories at different times is calculated relative to all cases at a site.*

8. Section 3.3: This section analyzes the consistency of different algorithms under various classification conditions, and finds out the reasons for the inconsistency. However, we notice that the inconsistency ratio under SBL classification is higher than that under other classification conditions. It is suggested to make a key explanation for the high inconsistency ratio under SBL classification.

*Response: Per your kind suggestion. We have further analyzed and explained the causes of the high inconsistency ratio of various algorithms under SBL classification in Section 3.3 of the revised manuscript.*